# A Brief Measure of Parental Wellbeing for Use in Evaluations of Family-Centred Interventions for Children with Developmental Disabilities

**DOI:** 10.3390/children7090120

**Published:** 2020-09-01

**Authors:** Roy McConkey

**Affiliations:** Institute of Nursing and Health Research, University of Ulster, Newtownabbey BT37 0QB, N. Ireland, UK; r.mcconkey@ulster.ac.uk

**Keywords:** parents, wellbeing, developmental disabilities, assessment, intervention, psychometric properties

## Abstract

Increasing emphasis is placed on the provision of family-centred interventions when children have developmental disabilities with the aim of supporting parents as well as fostering the child’s development. Although various instruments have been developed to assess parental health, stress and quality of life, these are rarely used by practitioners because of the burden they place on informants. A brief measure, rooted in the concept of subjective wellbeing, was developed and tested with over 400 parents of children with ASD participating in a home-based intervention. Consisting of eight items and using a 10-point rating scale, the measure was readily understood and accepted by parents. The items contributed to one main factor that confirmed the measure’s construct validity. The internal reliability of the scale was reasonable, and there was promising evidence of test–retest reliability. There is evidence too for criterion validity through a significant relationship with a measure of parental mental health. The summary score derived from the measure was sensitive to the predicted differences on wellbeing scores by parent characteristics as well as to features of their engagement with the intervention. This brief assessment tool could help practitioners to evidence the impact of their family-centred interventions.

## 1. Introduction

Parents of children with developmental disabilities—mothers especially—often report a lower quality of life [1], poorer physical and mental health [2], increased stress [3] and impaired family functioning [4]. To date, most of the interventions provided by teachers, therapists or psychologists are targeted at remediating the children’s behaviour or promoting their acquisition of skills. The needs of parents are often overlooked, as they also focus on the child rather than themselves [5]. Nevertheless, evidence is accumulating that interventions with children are more effective if delivered within a family-centred rather than child-focused approach [6]. Partnership working between practitioners and parents is recommended for several reasons. It facilitates greater consistency in how the child is managed at home as well as in the school or clinic; it provides the child with increased opportunities for additional practice and learning opportunities, and it encourages the generalization and consolidation of new skills and behaviours.

In recent years, however, the concept of partnership working has evolved to embrace the needs of parents and other family members [4]. A basic rationale for family-centred interventions is the mutual effects that parents and children have on one another. For example, a child’s challenging behaviours may result from, as well as contribute to, parental stress and depression [7]. Thus, practitioners aiming to promote children’s development need to attend to the wider needs of parents and siblings [5]. This new focus requires novel styles of working. This includes the development of family plans that include goals for the wider family as well as the child, the building of personal and trusted relationships between the parent and the key practitioner and basing the intervention within the family home rather than in schools or clinics [8]. 

New assessment tools are also required by practitioners wishing to monitor and evaluate changes in parents as a result of their interventions that can be used alongside the assessment instruments employed to evaluate the children’s progress. At present, many such tools are available which mostly take the form of questionnaires that parents can self-complete or the questions are asked during an interview with the practitioner [4]. However, many of the existing measures have been developed within the context of research and evaluation studies rather than with needs of practitioners in mind [9]. Practitioners require tools that can be quickly administered so as to reduce the burden on parents in both time and intrusiveness as well as ensuring that undue attention is not given to gathering information rather than on addressing the family’s intervention goals. Moreover, short questionnaires are more likely to be accessible to parents who have literacy or communication difficulties, and they are more easily translated into other languages [10]. 

Crucially, the questions in the questionnaire need to cover issues of concern to the target group of parents and the diversity of needs they may have [11]. These can be ascertained from studies that have consulted parents, by using questions commonly included in existing tools that have been used with the target group of parents and by trialling the items with groups of parents. The measure reported in the present study used all three methods over a period of some ten years evaluating the impact of different intervention programs with parents of children with Autism Spectrum Disorders (ASD) and intellectual disabilities [12,13,14]. From these experiences, the subjective wellbeing of parents emerged as the overarching framework which has been defined as: “an umbrella term for different valuations that people make regarding their lives, the events happening to them, and circumstances in which they live” [15]. Wellbeing embraced most of the issues that were pertinent to the parents with whom we worked and which the literature suggested had wider applicability, namely, physical and emotional health of parents, stress, coping, social support and quality of life [16]. 

The initial bank of items was refined and reduced as the agency practitioners used the Parent Wellbeing Measure in their work with families and children. Items on which there was minimal variation across parental responses were deleted. The final set of eight items is given in the Appendix A. A more visual and user-friendly rating scale was deliberately chosen rather than the usual five or seven-item Likert scales but with the inclusion of 10-point rating scale that would be more sensitive to detect changes in parental ratings over time.

But efficiency in administering questionnaires is not the sole consideration. The measure must be reliable and valid, especially when used to evaluate the impact of interventions on the parents. In this respect, some existing measures do not perform well [10]. Hence the aim of this paper is to report the psychometric properties of a brief measure of parental wellbeing when used with Irish parents, mostly of children with ASD and to establish its sensitivity to assessing change [17].

## 2. Materials and Methods

The parent wellbeing measure was administered as part of a five-year, brief home-based intervention project with 474 parents of children aged 2 to 11 years who had been diagnosed with ASD (87.2%) or were referred for an assessment because of suspected ASD (12.8%). The measures were taken at the start and at the end of intervention. The intervention was targeted at families who lived in designated geographical area with higher levels of social deprivation. Further details of the intervention are given in [14]. 

### 2.1. Participants

In all, 449 (95%) parents completed the parent wellbeing scale at the start and at the end of the home visits. Of these, 42 parents (9.4%) declined to give their demographic details. A summary of their child and family characteristics is given in Table 1. 

Most of the participating parents were female, aged under 40 years and were married. All but five (98.8%) declared as White British/Irish. As Table 1 shows, the sample included sizeable proportions of parents who were lone carers, who lived in families in which there was no wage earner, had left school at 16 years and lived in areas with high levels of social economic deprivation based on ratings provided by the Northern Ireland Statistics and Research Agency. Such parents are often less likely to take part in intervention projects [6].

### 2.2. Procedure

Parents were visited at home by project staff when they expressed an interest in participating in the study. Their informed consent to enrolling in the study included parents providing information about their child and family. Assurances of confidentiality were given, and parents were told that they could refuse to answer any question. 

The eight wellbeing items were presented on one A4 landscape sheet for the parents to self-complete. The items were expressed as polar opposites in two columns, separated by ten boxes from 1 to 10. Parents were instructed to select the box that best represented how they felt between the two extremes. Figure 1 is an example of one item. The final version of the scale is given in the Appendix A. 

The project officer read the instructions for rating the items and was available to answer any queries or to discuss parental responses. It usually took between two and four minutes for parents to make their responses. 

The information on parental wellbeing was gathered alongside the child and family demographics and further details of their child’s behaviours and the difficulties which the parents were experiencing. Learning goals were identified, and at a second and subsequent visits, an individual plan was devised to assist parents in meeting their goals. Parents were offered an average of five home visits over a 12-week period although the number could be increased or reduced depending on the child and family needs and preferences. In all, 174 parents (43.4%) received fewer than five visits, and 227 (56.6%) had five or more with a maximum of 12 visits. 

Once the family had received their allocation of visits, a final review session was held at which the parents were invited to re-assess their wellbeing on the eight items without any reference to their previous ratings. Again, this was in the context of further information being asked relating to the child’s progress and parents’ reactions to the project. 

A subset of parents were also invited to complete the Warwick–Edinburgh Mental Wellbeing scale [18] at the start and on completion of the home visits. This consists of 14 items that are rated on a five-point Likert scale that best describes the parents’ experiences of each item over the last two weeks. The scale ranges from “None of the time; Rarely; Some of the time; Often; All of the time”. The first three items are: “I’ve been feeling optimistic about the future; I’ve been feeling useful; I’ve been feeling relaxed”. A total score is computed by adding the ratings across the 14 items with higher scores indicative of better mental health. 

### 2.3. Data Analysis

All the item scores on the Wellbeing measures at the start and at the end of the home visits were entered into a SPSS spreadsheet along with the pertinent details of the parent and child characteristics. In order to maximize the number of respondents for statistical analyses, if parents had missed a response to one of the well-being items, a mean score on that item across all respondents was substituted using the missing values routine in SPSS. 

The following analyses were then conducted.

A principal components analysis was undertaken of the wellbeing items at the two time points in order to confirm that the same factor structure was evident in the parental responses. 

Cronbach alphas were calculated on the ratings given to the eight items at the two time points as a measure of internal reliability. This would give an indication of the measure’s construct validity. 

A total wellbeing score was calculated by adding the scores allocated across the eight items with higher scores indicative of better wellbeing. Pearson product moment correlations were calculated for scores at the start and at the end of home visits to provide an indication of test–retest reliability. The median period between the two administrations was eight weeks.

Similar correlations were also calculated between wellbeing scores and those attained in the Warwick Edinburgh scale in order to give an indication of criterion validity. 

Paired T-Tests were used to compare the wellbeing scores by the parental characteristics that past research had identified were associated with poorer wellbeing at both time points. Moreover, the impact on parental wellbeing scores of three variables descriptive of parental participation in the intervention were examined. They were the number of home visits undertaken, parent’s perceptions of the match between the number of sessions and their needs and the project officer’s rating of parental engagement with the intervention. Paired T-Tests on wellbeing ratings at the end of the home visits were used to identify significant differences. These analyses would provide an indication of the sensitivity of the measure in assessing change as well as offering further evidence of its criterion validity [17]. 

## 3. Results

The principal components analysis identified one main factor in parental ratings at the start of the home visits and again at the end. The total amount of variance explained was 47.5% for ratings at the start of the home visits and 52.8% at the end of the visits. 

Table 2 presents the factor loadings for the eight items along with the mean scores on each at the start and end of the project. On each item, the scores ranged from 1 to 10 with the scores approximately normally distributed on 14 of the 16 items in these analyses (skewness range from 0.01 to −1.36).

The Cronbach alpha across the ratings given to the eight items at the start was α = 0.827 and α = 0.861 at the end, which supports the internal reliability of the scale. 

A total score could be computed across the eight items by adding the numerical ratings given to each. The scores could range from 8 to 80. The mean score (and standard deviation) over all parents at the start was 50.29 (SD 13.30) with a skewness of −0.447 and for ratings at the end the mean score was 57.6 (SD 12.95) with a skewness of −0.700. The skewness values are indicative of normally distributed scores across the sample at both time points.

An estimate of test–retest reliability was obtained by calculating the correlation between total wellbeing scores for the same parents at the start and end of the home visits: r = 0.796 (*p* < 0.001). 

An indication of criterion validity was obtained for a subset of parents (*n* = 62) by correlating scores on the wellbeing scale with the ratings given to the Warwick–Edinburgh Mental Wellbeing Scale at the start of the home visits. The Pearson Product Moment correlation was r = 0.679 (*p* < 0.001).

The sensitivity of the wellbeing scale to assessing changes from participating in the intervention was demonstrated comparing the total scores at the start and after the intervention. Paired-Sample T-tests identified a statistically significant difference (*t* = −17.99: df 1:442: *p* < 0.001). The mean total score at the start was 50.29 (SD 13.30), and at the end it was 57.59 (SD 12.95), a difference of 7.3 points (95% Confidence interval 6.51 to 8.11). Table 2 confirms that on the eight items, parents scored higher at the end than at the start of the home visits (Paired *t*-tests *p* < 0.01). 

Further indications of sensitivity were examined in two ways. Past studies suggest that certain characteristics of parents are related to their wellbeing, and in this study, three proxy indicators were used, namely, lone parenting (lack of formal support), having no wage earner in the family (poverty) and age of parents [1]. Table 3 summarises the mean scores (and standard deviations) along with the results of statistical tests for these three variables.

One-way Analyses of variance were used to compare the differences within the subgroups in ratings made at the start and end of the home visits. As Table 3 shows, the differences in scores were statistically significant and in the predicted direction; lower wellbeing scores are found in families in which there is no wage earner, with lone parents and those who are aged 40 years and over. Moreover, the differences were replicated at the second administration of the wellbeing scale. 

A second indicator of sensitivity was the differential impact that the intervention may have had on parents’ wellbeing scores at the end of their involvement [13]. Three indicators were chosen: the number of visits families had received, whether the felt the number of sessions was appropriate for them, and the project staff ratings of the parents’ engagement with the intervention. As before, Table 4 summarises the mean scores (and standard deviations) along with the results of statistical tests for these three variables.

As Table 4 shows, there was no difference in mean wellbeing scores by the number of visits families had received, but there were significantly lower scores for parents who felt the sessions had been too short for their needs and for those whose engagement was rated by project staff as adequate or poor rather than good or very good.

Finally, a step-wise, linear regression was undertaken of parental wellbeing scores at the end of the home visits that examined all of the six variables described above, which were entered as two blocks as per Table 3 and Table 4. This analysis would account for the possible inter-relationships among these variables. The regression model was significant F = 13.86; df 3:384; *p* < 0.001 with R = 0.313. Three variables contributed significantly to the regression: no wage earner in the family (β = 4.81 SE 1.37; *p* < 0.001), number of visits too short (β = 6.92 SE 1.74; *p* < 0.001) and parental engagement judged as adequate or poor (β = 4.72 SE 1.54: *p* < 0.002).

## 4. Discussion

This brief measure of parental wellbeing was well received by parents and has promising psychometric properties. The items were chosen for their face validity and relevance to parents of children with developmental disabilities. The items contributed to one main factor of wellbeing that confirmed the measure’s construct validity. There is evidence too for criterion validity through the significant relationship with a measure of parental mental health. Moreover the internal reliability of the scale is reasonable, and there is promising evidence of test–retest reliability albeit this was based on measures taken at the start and at the end of the home visits covering a median time period of eight weeks and when there may have been a differential impact of the intervention, all of which may have reduced the test–retest correlation. The summary score from the measure was sensitive to the predicted differences on parental wellbeing, which previous studies had reported as well as to features of their engagement with the intervention.

Further studies could usefully assess the utility of the measure with families of children with other intellectual or physical disabilities, or those with chronic illnesses. Although parental gender differences were not found in this study, this might be tested more thoroughly by comparing ratings given by males and female caregivers of the same child. The use of the scale with parents in other countries would help to determine its cultural validity.

The tool is also intended for practitioners to use in their clinical practice. The pattern of responses which parents give to the items could trigger discussions with the parent about their personal wellbeing and the changes they would like to bring about. From these discussions, personal goals might be devised and action plans made for achieving them. This could include activities known to improve parental wellbeing such as mindfulness [19] as well as those aimed at improving parent-child interactions and changing the child’s behaviours [9]. Having the parent re-rate themselves could give an indication of progress both for their own benefit as well as that of the practitioners wanting to gather evidence of the effectiveness of the practice [8].

## 5. Conclusions

This brief assessment tool could help to bolster evidence-based, family-centred interventions for children with developmental disabilities which are often recommended but as yet have proven difficult to realize.

## Figures and Tables

**Figure 1 children-07-00120-f001:**
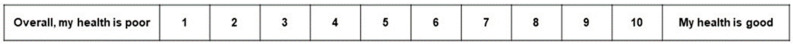
An example of the layout of the questionnaire.

**Table 1 children-07-00120-t001:** The characteristics of the participants.

**Characteristics of Children with ASD**		**Number (%)**
Gender	Male	334 (74.4%)
	Female	115 (25.6%)
Age	Less than 6 years	162 (40.1%)
	6 years and over	242 (59.9%)
School attended	Special School/Unit	47 (11.5%)
	Mainstream school	401 (88.5%)
**Characteristics of Parent**		**Number (%)**
Gender	Female	377 (92.6%)
	Male	30 (7.4%)
Age	Under 40 years	290 (71.4%)
	40 years and over	116 (28.6%)
Marital Status	Single/Divorced/Widowed	135 (33.2%)
	Married	272 (66.8%)
Wage earner in family	No	128 (31.5%)
	Yes	278 (68.5%)
Education	Left school 16	189 (46.6%)
	Further and Higher Education	217 (53.4%)
Level of deprivation	Most deprived areas	115 (26.8%)
	Less deprived areas	113 (26.3%)
	Least deprived areas	201 (46.9%)

**Table 2 children-07-00120-t002:** The factor loadings and the mean and standard deviations (SD) of the scores of each item at the start and end of the home visits. (*n* = 429).

Items	Factor Loading Start	Factor Loadings End	Mean Start	SD Start	Mean End	SD End
Feeling Down–Great	0.808	0.833	5.98	2.06	6.99	1.97
Poor–Good Quality of Life	0.796	0.792	7.43	2.05	8.07	1.88
Stressed–Relaxed	0.708	0.763	4.75	2.17	6.17	2.16
Hard–Easy: Managing Household Tasks	0.692	0.746	6.88	2.51	7.75	2.16
Time in House–Go out of House	0.662	0.716	5.73	2.97	6.67	2.86
Poor–Good Health	0.616	0.652	6.94	2.28	7.61	1.99
Difficult–Easy: Looking After Child	0.609	0.636	6.12	2.42	7.27	2.09
No friends–Friends	0.592	0.648	6.46	3.07	7.05	2.85

**Table 3 children-07-00120-t003:** The mean and standard deviations (SD) of the wellbeing scores for the subgroupings of parental characteristics at the start and end of the home visits. (*n* = 406).

Parental Characteristics	Mean Ratings (SDs) at Start	Mean Ratings (SDs) at End
No wage earner in family	47.28 (14.30)	54.00 (13.62)
Wage earner in family	52.21 (12.67)	59.84 (12.19)
Statistical Tests	F = 12.19: *p* < 0.001	F = 18.67: *p* < 0.001
Lone parent	47.88 (14.12)	55.84 (13.29)
Both parents	51.94 (12.72)	59.05 (12.63)
Statistical Tests	F = 8.49: *p* < 0.005	F = 5.62: *p* < 0.05
Parent under 40 years	51.75 (12.09)	58.84 (11.89)
Parent 40 years and over	47.79 (15.81)	56.03 (15.07)
Statistical Tests	F = 7.39: *p* < 0.01	F = 3.92: *p* < 0.05

**Table 4 children-07-00120-t004:** The mean and standard deviations (SD) of the wellbeing scores for the subgroupings of parents by their engagement with the project at the start and end of the home visits. (*n* = 406).

Parent Engagement	Mean Scores (SDs) at End
Up to 4 home visits	58.81 (13.78)
5+ home visits	57.47 (12.45)
Statistical Tests	F = 1.0419: NS
Number of sessions too short	52.06 (12.68)
Number of sessions “just right”	58.87 (12.68)
Statistical Tests	F = 15.97: *p* < 0.001
Good co-operation	59.05 (12.89)
Adequate and poor cooperation	54.00 (12.82)
Statistical Tests	F = 10.65: *p* < 0.01

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
