# Peer review of "A Brief Measure of Parental Wellbeing for Use in Evaluations of Family-Centred Interventions for Children with Developmental Disabilities"

_children, 2020, doi:10.3390/children7090120_

Round 1

Reviewer 1 Report

This paper reports psychometric evaluation of a newly developed short instrument (eight items) measuring parental wellbeing, especially among children having developmental disabilities. This kind of short instrument is needed and it is most probably useful in other pediatric contexts as well.  

The sample is sufficient (n=449) to make this kind of evaluation of psychometric properties. The sample was gathered at two time points which also strengthens the design and makes the validation stronger. The psychometric properties are tested with several adequate methods, such as PCA (construct validity), criterion validity by correlations, and test-retest reliability tests (correlation). Additionally the instrument seems to be quite sensitive to the predicted differences on wellbeing scores by parental background factors. The manuscript is logically and well written and it is smooth to read.  

There are minor typo errors which should be corrected. E.g. tables 3 and 4 the extra parenthesis  

Statistical Tests F=7.39: p<0.01)       F-3.92: p<0.05)

Author Response

The typos identified by the reviewer in Tables 3 and 4 have been corrected. 

Reviewer 2 Report

Thank you for inviting me to review: " Brief Measure of Parental Wellbeing for Use in Evaluations of Family-Centred Interventions for Children with Developmental Disabilities".

The article approaches an important and under-represented area of study: family well-being and the effects of interventions for children with Intellectual Disabilities and Autism Spectrum Disorders on their families. Availability of instruments for practitioners and for research is scarce, for this reason the article provides an important contribution to the field.

The article is well written and the methodology is sound. It provides basic information regarding the validity and reliability of the instrument. Further clarifications  may clarify highlight the work that has been done to create the instrument.

Page 2 line 72: Please further clarify the methodology chosen for the item reduction from the bank of items. A further explication would increase the face validity of the instrument.

Page 3 line 94 please rephrase: “However the sample included…” Was the sample representative of service users? Was the stress on the social frailty correlated to the white majority of the population?

Page 3 line 100: Were any ethical standard used? Please specify.

Page 4 Line 144 What was the time span between the test and re-test?

Reference List: Some of the author references are not completed.

Author Response

Page 2 line 72: Please further clarify the methodology chosen for the item reduction from the bank of items. A further explication would increase the face validity of the instrument.

This sentence has been added:  Items on which there was minimal  variation across parental responses were deleted.

Page 3 line 94 please rephrase: “However the sample included…” Was the sample representative of service users? Was the stress on the social frailty correlated to the white majority of the population?

The sentence has been rephrased.  "As table 1 shows, the sample included ...."

This makes clear that socially frailty was predominantly within the white majority population as only five persons declared as non-White. 

Page 3 line 100: Were any ethical standard used? Please specify.

This information has been added: "Their informed consent to enrolling in the study included parents providing information about their child and family. Assurances of confidentiality were given and parents were told that they could refuse to answer any question". 

Page 4 Line 144 What was the time span between the test and re-test?

This sentence has been added.  The median period between the two administrations was eight weeks.

Reference List: Some of the author references are not completed.

The authors references are now given in full. However one paper has yet to be accepted for publication but a preprint is available on ResearchGate